# The Hungarian hubris syndrome

**Lilla Magyari**[1,2]*, **Csaba Pléh**[3], **Bálint Forgács**[4]

**1** Department of Social Studies, Faculty of Social Sciences, University of Stavanger, Stavanger, Norway, **2** Norwegian Centre for Reading Education and Research, Faculty of Arts and Education, University of Stavanger, Stavanger, Norway, **3** Department of Cognitive Science, Central European University, Budapest, Hungary, **4** Department of Cognitive Psychology, Institute of Psychology, ELTE Eötvös Loránd University, Budapest, Hungary

* lilla.magyari@uis.no

## Abstract

Powerful figures, such as politicians, who show a behavioural pattern of exuberant self-confidence, recklessness, and contempt for others may be the subject of the acquired personality disorder, the *hubris syndrome*, which has been demonstrated to leave its mark on speech patterns. Our study explores characteristic language patterns of Hungarian prime ministers (PMs) with a special emphasis on one of the key indicators of hubris, the shift from the first person "I" to "we" in spontaneous speech. We analyzed the ratio of the first-person singular ("I") and plural ("we") pronouns and verbal inflections in the spontaneous parliamentary speeches of four Hungarian PMs between 1998–2018. We found that Viktor Orbán during his second premiership (2010–2014) used first person plural relative to singular inflections more often than the other three PMs during their terms. Orbán and another Hungarian PM, Ferenc Gyurcsány, who were re-elected at some point showed an increased ratio of first-person plural vs. singular inflections and personal pronouns by their second term, likely reflecting increasing hubristic tendencies. The results show that the ratio of "I" and "we" usually studied in English texts also show changes in a structurally different language, Hungarian. This finding suggests that it is extended periods of premiership that may increase hubristic behaviour in political leaders, not only experiencing excessive power. The results are particularly elucidating regarding the role of re-elections in political leaders' hubristic speech–and behaviour.

## Introduction

Over a decade ago David Owen and Jonathan Davidson [1] proposed a new kind of personality disorder, which may be acquired by successful political leaders exactly because of their success: the hubris syndrome (HS). Since then the idea has emerged that particular speech features of political and business leaders may not only report how spoiled they are by power, but could be used as linguistic biomarkers with diagnostic value [1–6]. The present paper sets out to identify individuals affected by HS among Hungarian prime ministers (PMs) based on the linguistic markers expressed in their spontaneous speech.

The overconfidence and sense of invulnerability observed in leadership-behaviour, described originally in behavioural finance as *hubris hypothesis* [7], was suggested to lead to

**Data Availability Statement:** All raw data is publicly available on the official website of the Hungarian Parliament (kormany.hu, 2021) (https:// www.parlament.hu). Data selected for the analyses are available here: https://osf.io/4tkvj/.

**Competing interests:** The authors have declared that no competing interests exist.

disadvantageous decisions and misjudgements or even unethical behaviour [8–11], even though some positive consequences have been pointed out recently [12]. Hubristic behaviour of political leaders has got into the spotlight with the recent rise of populism [13, 14]. Based on the behavioural pattern and medical history of politicians, being spoiled by power has been conceptualized as an acquired personality disorder due to a sustained experience of excessive power [1, 2]. Owen and Davidson [1] described HS in medical terms as a disorder with fourteen diagnostic criteria including verbal behaviour, speaking style and word choices. However, HS might be more akin to a personality trait rather than a personality disorder, since it usually appears later in life as a consequence of and during holding a powerful position [15].

In their landmark study, Garrard et al. [3] provided a detailed linguistic analysis of the parliamentary speeches of UK PMs and identified speech patterns linked to HS. Both Margaret Thatcher and Tony Blair showed a higher ratio of using the first-person plural pronoun, "we" relative to its singular pair, "I", during periods of their governance, when they expressed multiple additional behavioral markers of HS. Ensuing studies used text analysis tools to study the hubristic language use of CEOs [4, 16, 17]. Akstinaite et al. [17] confirmed that personal pronouns, and specifically the use of "we", is a strong and diagnostic marker of hubristic language; whereas narcissism, a closely related personality disorder, has been found to correlate with the second person pronoun "you" [18], and not with "we". In sum, the high proportion of "we" in natural speech seems to be a powerful linguistic biomarker of hubristic personality traits with differential diagnostic value. For this reason, we decided to focus our inquiry on this particular linguistic feature.

While hubristic behaviour can lead to low quality decision making with a high risk of misjudgement, only a few quantitative studies used text-analysis methods to evaluate HS in political leaders [3, 19]. While linguistic markers of HS have been described exclusively in English so far [3, 17, 19], we aimed to analyze political speeches of PMs in a non-Indo-European language, Hungarian in our study.

Among prime ministers of Hungary, Orbán has been in power for the longest period since the fall of the iron curtain, between 1998 and 2002 and since 2010 to this day, which makes him particularly vulnerable to HS. It has also been suggested that subsequent re-elections might increase the chance for hubris [15]. Orbán's political behaviour and leadership style from his second term onwards (since 2010) is consistent with this idea, even though it is also true that when re-elected, he obtained a supermajority, which provided him excessive power. Since then, he is renowned for limiting challenges to his power by tampering with electoral laws and the constitution, constraining free speech and independent media, and eroding civil and human rights in Hungary [14, 20]. In the past decade he has been regarded as an authoritarian populist leader at the far right of the political spectrum [14], on par with Erdogan in Turkey or Putin in Russia.

The particularities of Hungarian language allow for specific predictions on hubristic linguistic markers. In Hungarian, subject pronouns are regularly dropped in sentences, with number and person expressed by verb conjugation. Personal pronouns can also be included in sentences, but they are utilized usually for emphasis, such that, most personal pronouns in subject roles are contrastively focused [21]. Thus, not only the usage patterns of first-person pronouns but also verbal inflections are highly informative regarding speech patterns in Hungarian.

For our investigation we sought for official but spontaneous political speeches. To this end, we selected speeches from the official records of the Hungarian parliament that were likely not prepared in advance. Our hypotheses are the following: Orbán, due to hubristic personality traits uses the personal pronoun "we" in a higher proportion relative to "I" (increased WE:I ratio) in his speech than other former Hungarian PMs, especially from his second premiership

onwards, when he obtained a supermajority and has experienced excessive power ever since. It has been suggested that subsequent re-elections might increase hubristic tendencies [15], therefore, we also hypothesized that both Orbán and Gyurcsány may show increased linguistic markers of hubris after being re-elected. We expect Orbán's WE:I ratio to remain consistent during his years in power since 2010, as hubris, once triggered, should be present as long as its triggering conditions are met (he retained his supermajority in all ensuing elections). Further on, as verbal suffixes carry number and person marking in Hungarian, we also predict that the WE:I ratio effects will be present in first person verb conjugations as well.

## Materials and methods

### Subjects

Speeches of all Hungarian PMs were studied who governed for various durations between 1998–2018. Parliamentary elections are held every four years in Hungary, but the ruling party may switch PMs, which led to seven governments headed by four PMs during this period. Two out of the four PMs, Orbán and Gyurcsány, held office more than once (Fig 1, and see also Subjects in S1 File).

### Speech samples

Similarly to Garrard et al. [3], transcripts of public speeches on plenary sessions of the Hungarian Parliament were obtained [22]. Instead of analysing spoken utterances of interviews (e.g., see [17]), we used parliamentary speeches because this allowed us to compare spoken utterances arising from a similar context and situation among different prime ministers. The parliamentary records from 1998 include a classification system which helped us to select those speeches which are the most spontaneous and to avoid prepared speeches which might be written by professional speech writers. We chose the following categories of speeches of all politicians who were prime ministers between 1998 and 2018 in Hungary: 1) *two-minute interventions*: a remark during general debate with permission from the parliament chair. 2) *reply*: reply by the speaker during general debate; 3) *re-replies*: one-minute re-replies to replies by speakers to so called immediate questions. Altogether 454 replies and interventions were selected from the four PMs through seven parliamentary cycles from July 6th, 1998 to May 7th, 2018. All selected speech samples were parsed for grammatical structure [23]. Using the grammatical parser, we retrieved personal pronouns (PP) with their variations (i.e., in different cases, see [24]). We also retrieved verbs with first person singular and plural verbal inflections (verb conjugations, VC) for all samples. The form of the nominative case of the first-person

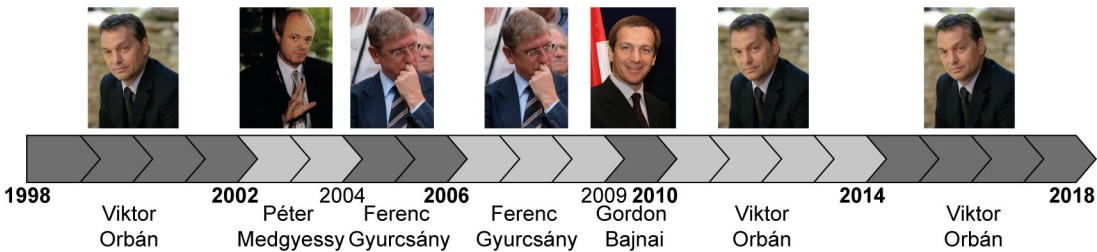

**Fig 1. Timeline of parliamentary cycles and prime ministers in Hungary from 1998 to 2018.** The bottom line shows the name of the PMs and years of change in PM and/or the years of parliamentary elections (the later are in bold). All images are downloaded from Wikimedia Commons (commons.wikimedia.org; Orban Viktor Portrait.jpg, kormany.hu, 2010, CC BY-SA 2.5; Medgyessy Péter 2002.jpg, Tamás Griechisch, 2002, CC BY-SA 4.0; Gyurcsány and Medgyessy (Aug 2014).jpg, Tibor Végh, 2014, CC BY 3.0; Bajnai Jerusalem.jpg, Itzike, derivative work by Qorilla, 2009, CC BY-SA 3.0).

plural pronoun, "mi" in Hungarian ("we" in English) is the same as a less frequent variation of the relative pronoun "what". Therefore, a research assistant checked all speech samples manually and corrected in the data table if the word "mi" was wrongly categorized by the analyser, and the lead author of the study checked again if the manual corrections were well done.

## Statistical analyses

The amount of PP and VC within speech samples and the number of speech samples per PM and parliamentary cycles varied considerably, therefore, we used generalized linear mixed-effect model (GLMM) analysis to control for the varying occurrences of these. Binomial GLMMs were fitted on two categorical response variables, 1) on the singular and plural first-person personal pronouns (PP), i.e. "I" and "we" and their variations (singular = 1, plural = 2) and 2) on the singular and plural first person verb conjugations (VC) (singular = 1, plural = 2), using the glmer() function of the lme4 package in R [25]. We included speech sample as a random factor in our models. Fixed effects were the PMs and/or their terms of premiership. Models with more than one fixed effect included an interaction term only if it significantly improved them. The significance of interaction between fixed effects and the significance of the fixed effects were evaluated by model comparison with likelihood ratio test. When the GLMM contained one significant fixed effect with more than two levels, the model was run repeatedly, each time with a different reference-category for the fixed effect. The coefficients of the categories of fixed effects revealed whether they are different from the coefficient of the intercept (i.e., from the reference-category). This way, we could also test the differences between each of the levels of the fixed effect.

Model diagnostics were applied using the DHARMa package [26]. It used a simulation-based approach to create readily interpretable scaled (quantile) residuals for fitted (generalized) linear mixed models. If the model diagnostics returned any errors with the model, we excluded outlier speech samples that contributed extremely high amount of data (more than 50 items) in order to reduce variation between samples. Then, the GLMM analysis was run again on the newly reduced dataset.

## Results

### Personal pronouns

In the speech samples, 2.9% of all words were personal pronouns. 60.2% of all these pronouns were in nominative case, 14.5% were in dative case, 10.7% were in accusative case, 3.8% in instrumental case, 2.7% in sublative case, 2% in inessive case, 1.8% in ablative case [24]. All other forms were under 1%. Following Tyrkkö's analysis of pronouns in political speeches [27], we also categorized the personal pronouns according to semantic categories based on their referents. The first-person singular pronouns ("I") and their variants were considered Self-referential, the second person singular and plural ("you") and their variants were Audience-referential, and the third-personal singular and plural ("he/she/it" and "they") and their variants were Other-referential. Tyrkkö categorized the first-person plural pronouns ("we") as Inclusive-referential assuming that the majority of such pronouns are delivered in an inclusive sense in political speeches. We do not make such assumption; hence, we just call this category "WE". Fig 2 shows the relative frequency of categories of the personal pronouns in our samples. The Audience-referential pronouns have the overall highest frequency in our data. Audience-referential pronouns are also the most frequent category in each parliamentary cycle for each PM, except of Bajnai who has the same amount of Audience and Self-referential personal pronouns (S1 Table in S1 File).

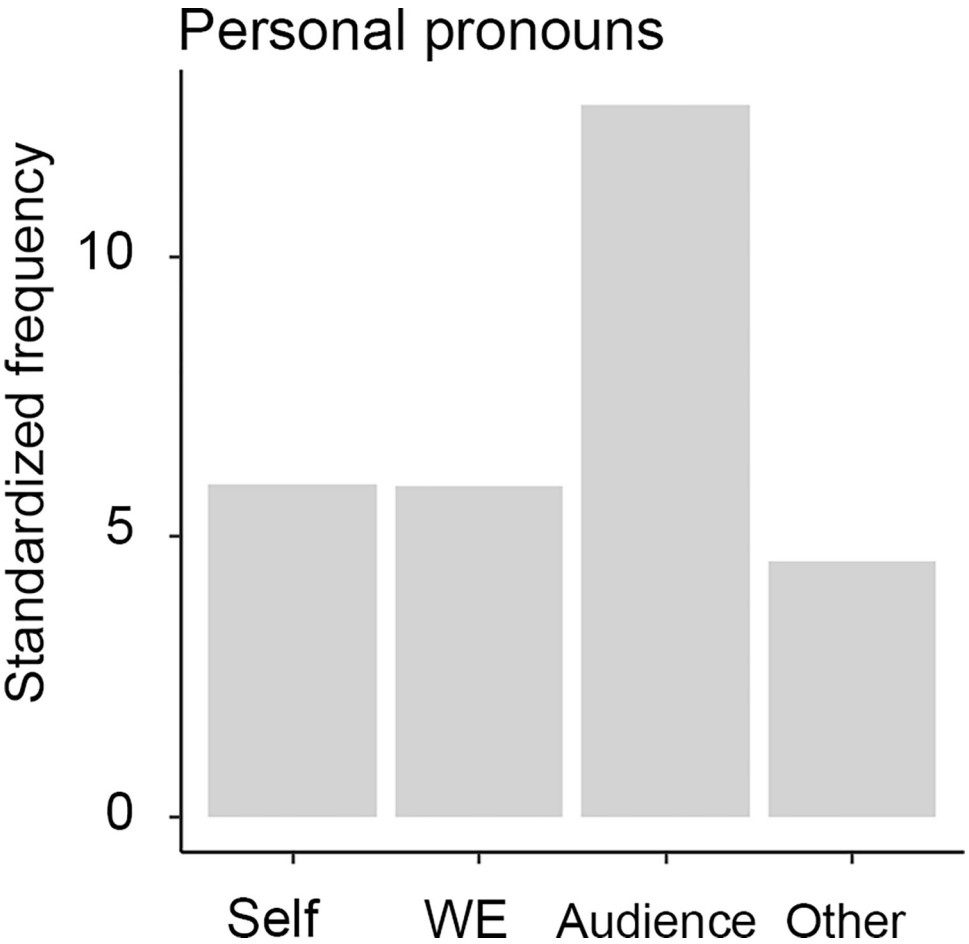

**Fig 2. Frequency of personal pronouns.** Standardized frequency (number of pronouns/1000 words) are shown on the y-axis for each category of pronouns (x-axis) in all speech samples. Self: first-person singular pronouns ("I"), WE: first person plural pronouns ("we"), Audience: second-person singular and plural pronouns ("you"), Other: third person singular and plural pronouns ("he/she/it", "they").

The number of first-person PPs and VCs and speech samples used for data analysis are shown in Table 1.

First, we compared the ratio of singular and plural first-person PP (WE:I ratio) among the four PMs (Fig 3A).

The number of all speech samples and the number of PPs ("we" & "I") were very different across PMs. Orbán had largest number of items and speech samples. However, we hypothesized that Orbán acquired hubristic traits from his second premiership (between 2010–2014). Therefore, we used speech samples of one single term for each PM and we selected the second term of premiership for Orbán and for Gyurcsány (Table 1).

The binomial GLMM showed that the WE:I ratio in PP significantly varied by PM, $X^2(3) = 10.33$, $p = .016$; Intercept (reference category: Orbán), $\beta = 0.288$, SE = 0.196, $z = 1.467$, $p = .142$; Fig 4A. There were less plurals ("we") in PP for Medgyessy compared to Orbán ($\beta = -1.219$, SE = 0.42, $z = -2.90$, $p = .004$). The difference was marginal between Orbán and Bajnai ($\beta = -0.858$, SE = 0.469, $z = -1.83$, $p = .067$). There were no other differences between PMs.

We also tested whether there was a difference between the first and second premiership for Orbán and Gyurcsány (Table 1). There was no difference between the two PMs but there were

**Table 1. Frequencies of first-person personal pronouns and verb conjugations.**

| Prime ministers | First-person personal pronouns | | | First-person verb conjugations | | |
|---|---|---|---|---|---|---|
| | n[a] | %[b] | n (s.s.)[c] | n | % | n (s.s.) |
| P. Medgyessy (2002–2004) | 58 | 31 | 30 | 436 | 32 | 59 |
| F. Gyurcsány | | | | | | |
| -First (2004–2006) | 26 | 23 | 15 | 210 | 30 | 32 |
| -Second (2006–2009) | 84 | 54 | 12 | 317 | 45 | 17 |
| G. Bajnai (2009–2010) | 42 | 38 | 17 | 228 | 36 | 27 |
| V. Orbán | | | | | | |
| -First (1998–2002) | 57 | 42 | 26 | 379 | 32 | 53 |
| -Second (2010–2014) | 203 | 57 | 73 | 875 | 46 | 101 |
| -Third (2014–2018) | 229 | 54 | 92 | 954 | 40 | 137 |

Data is shown for the four Hungarian PM from 1998 to 2018.

[a]Number of items.

[b]Percentage of plurals.

[c]Number of speech samples.

more plurals in their second term compared to their first premiership, $X^2(3) = 10.33$, $p = .016$, $\beta = 1.09$, SE $= 0.369$, $z = 2.953$, $p = .003$. However, there was no difference in the WE:I ratio for PP between Orbán's second and third term, $X^2(1) = .249$, $p = .6181$, $\beta = -0.138$, SE $= .276$, $z = -0.499$, $p = .618$, Table 1. Model diagnostics of these analyses are shown in S1 Fig in S1 File.

An additional analysis of the personal pronouns can be found in the Supporting Information (Supplementary Results and Supplementary Discussion). The analysis compares the changes in the frequency of the first- and third-person plurals relative to the other pronouns in Orbán's and Gyurcsány's speeches in their first and second term as PMs.

## Verb conjugations

15% of all words were verbs. We did not categorize the inflected variants of verbs into semantic categories of its substantives because the third-person verbal conjugations were overwhelmingly used for second-person subjects, "you". Although Hungarian singular and plural first, second and third-person verbal conjugations can be differentiated based on their forms, third-person is used instead of the second-person verb conjugations in polite, distant or official interactions (so called, V-forms in language [28]) [29]. In our samples, all Audience-related pronouns were expressed in their V-form variants which shows that PMs' language use overwhelmingly followed the polite forms of interactions.

The WE:I ratio for verb conjugations were also compared across PMs (Fig 3B). For Orbán and for Gyurcsány, data from their second premiership was entered into this analysis (Table 1). The binomial GLMM showed that WE:I ratio for VC, i.e., the first-person verb conjugation being singular or plural, was significantly influenced by PM, $X^2(3) = 19.01$, $p = .0006$; Intercept (reference category: Orbán), $\beta = -0.259$, SE $= 0.117$, $z = -2.204$, $p = .0275$; Fig 4B. There was a higher ratio of WE:I for VC for Orbán compared to Bajnai ($\beta = 0.54$, SE $= 0.255$, $z = 2.113$, $p = .004$) and compared to Medgyessy ($\beta = 0.815$, SE $= 0.211$, $z = 3.863$, $p = .0001$). The difference was marginally significant between Orbán and Gyurcsány ($\beta = 0.571$, SE $= 0.301$, $z = 1.899$, $p = .058$; Fig 4B). However, in an outlier filtered analysis, when some speech samples were excluded from the data due to poor model diagnostics, the statistically improved model showed significantly more plurals for Orbán relative to Gyurcsány (see S2 and S3 Tables in S1 File and S2 Fig in S1 File).

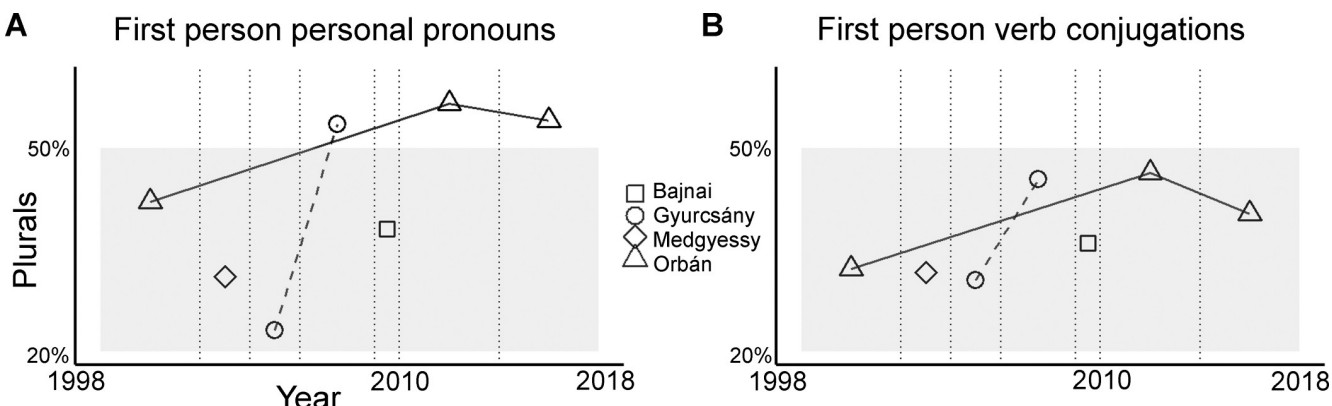

**Fig 3. Percentage of plurals (WE).** Percentages are shown A) within all first-person personal pronouns ("I" and "we"), and B) within first-person verb conjugations. Data shown for each parliamentary cycle (vertical dotted lines on y-axis) and for each PM (symbols).

There was also a difference in the WE:I ratio for VC between Orbán's and Gyurcsány's terms (first vs second), $X^2(1) = 16.84$, $p < .0001$, and also between persons, $X^2(1) = 7.17$, p = .007; Table 1. The number of plural forms increased in the second term compared to their first term ($\beta = 0.668$, SE = 0.162, $z = 4.12$, $p < .0001$), while Gyurcsány had fewer plural inflections compared to Orbán, $\beta = -0.507$, SE = 0.193, $z = -2.63$, $p < .001$. When outlier speech samples were rejected to improve model diagnostics (S3 Fig in S1 File), the analysis of the reduced data-set showed similar results (see S2 and S4 Tables in S1 File S2, S4 and S3 Figs in S1 File). In his third term, Orbán's data showed a marginally lower ratio of plurals compared to his second term, $X^2(1) = 3.77$, $p = .0521$; $\beta = -0.27$, SE = 0.139, $z = -1.947$, $p = .052$. An improved model showed similar results (see S2 and S5 Tables in S1 File and S4 Fig in S1 File).

A tentative analysis of speeches of Gyurcsány and Orbán as a Member of Parliament and not as PM can be found in Supplementary Results and Supplementary Discussion of the Supporting information.

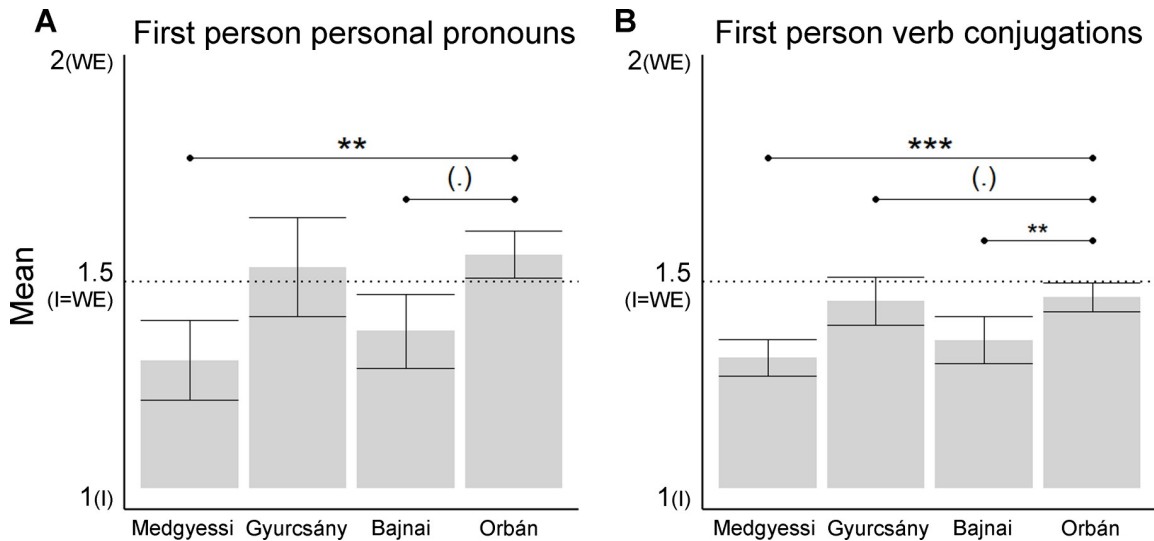

**Fig 4. Mean of the categorical variable coding the first-person plurals and singulars.** Means are shown A) for personal pronouns and B) for verb conjugations for each Hungarian PMs (x-axis). On the y-axis 1 indicates the exclusive use of singulars ("I"), while 2 indicates the exclusive use of plurals ("we"). For Gyurcsány and Orbán, the second term of their premiership is plotted. Error bars show confidence intervals, horizontal lines show significant differences and trends between PMs. (.): $p < .1$. *: $p < .05$. **: $p < .01$. ***: $p < .001$.

# Discussion

We investigated the speech patterns of Hungarian PMs in order to establish whether the current PM, Viktor Orbán's speech exhibits the key linguistic biomarker of hubristic personality trait, that is, the excessive use of the personal pronoun "we" instead of its first-person counterpart "I". Our study also extends earlier research of the linguistic biomarkers (e.g., [5, 17, 19]), because the Hungarian language also allowed us to examine the WE:I ratio in terms of verb conjugations besides personal pronouns. We found that Orbán had a higher WE:I ratio both for personal pronouns and for verb conjugations in his second compared to his first premiership. Moreover, in his second term, Orbán's WE:I ratio for verb conjugations was higher than any other Hungarian PMs'. Another PM, Gyurcsány, who also governed for more than one parliamentary cycle, also had higher WE:I for personal pronouns and for verb conjugations in his second compared to his first term.

Earlier works found personal pronouns to be a reliable marker of personality traits and social hierarchy [5, 17, 30–32]. However, most of these studies used computerized text analysis of English texts or English translation of texts written in other language (e.g., [31]). Here, our results showed that the relative frequency of first-person personal pronouns can be still a relevant linguistic marker of hubris even in a non-Indo-European language, like Hungarian, in which the frequency and linguistic behaviour of pronouns are governed by to different rules compared to English. Moreover, first-person verb conjugations in Hungarian might also fulfil similar diagnostic role as personal pronouns in identifying hubristic behaviour. In our study, we found that the ratio of the plural first person pronouns and verb inflections showed almost similar results. Therefore, future research could further study whether there are the differences in the function of the two types of linguistic forms as markers of hubris.

The increase between the first and second terms for both Orbán and Gyurcsány is in line with description of the hubristic personality that states that hubris tends to develop over time following the acquisition of power. For example, Garrard et al. (2014) showed that Margaret Thatcher's usage of "we" compared to "I" peaked in the year of her re-election. But other speech corpus data confirming the effect of re-election is scarce. Orbán's WE:I ratio did not increase from his second to his third term, which suggests that the effect of hubris on these linguistic markers remained at its peak. The first re-election might be of particular importance. Crucially, we also found that Gyurcsány's speeches also exhibited a higher WE:I ratio in his second compared to his first term. Although we could show the effect of re-election only for two (male) PMs due to our restricted sample size, these results suggests that re-elections may have a general effect of increasing hubristic tendencies in political leaders. Future research could target especially, the effect of re-election on hubris for more PMs and for more individual variation (for example, such as variation in gender and in political-geographical context).

In general, high status individuals use a higher WE:I ratio that may reflect that they are more collectively minded or other-oriented compared to lower ranking persons [31]. Hence, frequent usage of "we" might not be a simply hubristic marker after all. However, it has been shown that higher frequency of "we" differentiates higher ranked people with and without hubristic personality [3, 17]. In our study, we also found that Orbán's WE:I ratio in verb conjugations was higher compared to all other PMs, a control group of similarly high-ranking politicians.

In our analysis we focus on the changes in the relative ratio of the singular and plural personal pronouns (WE:I). While the frequent usage of certain pronouns could be also related to other personality traits (e.g. narcissism [18]), to rhetorical strategies (e.g., [27, 33]) or to the communication of ideologies (e.g., [34]), the relative increase of the plural first-person

personal pronouns compared to its singular form have been exclusively described as a correlate of hubristic behavior ([5, 19]).

For example, the plural first-person pronouns ("we", "us") was found to be the most frequent among all other pronouns in pre-written political speeches [27, 35]. The frequent use of "we" and "us" in such speeches is assumed to be a rhetorical strategy to foster a sense of community with audience [27]. In this study, we analyzed remarks, replies and re-replies of parliamentary speeches which were probably not prepared in advance. In contrast with pre-written political speeches, we found that the Audience-referential (i.e., second-person plural and singular) were the most frequent pronouns in our sample (Fig 2, S1 Table in S1 File). We cannot exclude the possibility that such a pattern is specific to Hungarian because quantitative studies of political speeches typically rely on English texts. However, in our view, it is more likely that Audience-related pronouns had the highest frequency in our study, exactly because the speeches were uttered in the heat of political debates directed towards specific communicational partners spontaneously. Such a finding further corroborates the spontaneity of the selected speech samples.

The WE:I ratio might change in spontaneous speech because it is especially a unique hubristic trait to use "we" where its singular form would be suitable just as well. Two of the symptoms of hubris refer to such language use: 1) when one uses the "royal we" and 2) when own interests or desires are identified with a nation's (or organization's) needs [1, 2]. There are such typical examples of the usage of "we" in Orbán's speeches when he speaks not only in the name of his party or the government but in the name of his voters. An example of this when Orbán said that "we, the Hungarians of the 21st century could accomplish our own revolution based on the April election" (kormany.hu, on 29.05.2010, 2nd speech). This illustrates that Orbán identifies his opinions and actions with those of his audience and talks to his voters as if they were the Hungarian nation as a whole. Future qualitative studies would be necessary to reveal the pragmatic function and referent of "we" in political speeches and whether there is a change also in their function accompanying the change in the WE:I ratio.

Politicians with no or few constraints on their decisions and behaviour, such as authoritarian leaders are particularly sensitive for acquiring hubris [1]. Viktor Orbán, the current PM of Hungary emerged into a powerful position during his second premiership in 2010 when his party obtained a two-thirds majority of the Parliamentary seats. This enabled Orbán to eliminate constraints on his power: he has been criticized world-wide since then as a leader who abuses his power to build an authoritarian, "illiberal" democracy [14]. Here we showed that linguistic markers of hubris unique to and typical for HS have become more frequent in Orbán's semi-spontaneous parliamentary speeches exactly in this period of unlimited political "success".

Hubristic traits in leaders might lead to low quality decision making and potentially unethical behaviour, hence, it is particularly important to identify hubristic traits in politicians' behaviour for which linguistic markers may provide a powerful tool.

Our results suggest that in Eastern-Europe, a region with limited history with liberal, parliamentary democratic norms and governance, hubris emerges and increases in power positions, just as in Western democracies. In other words, the dynamics of democratic deterioration may not be reversed in less developed countries, such that power hungry strongmen emerge to power, but democratic institutions may resist less when leadership success makes leaders power hungry and authoritarian.

## Supporting information

**S1 File.**
(PDF)

## Acknowledgments

We are grateful to Gábor Prószéky for his help with the grammatical parser and to Nguyen-Dang Nóra Lien for her enormous help with the organization of the speech corpus.

## Author Contributions

**Conceptualization:** Lilla Magyari, Csaba Pléh, Bálint Forgács.

**Data curation:** Lilla Magyari.

**Formal analysis:** Lilla Magyari.

**Methodology:** Lilla Magyari, Csaba Pléh, Bálint Forgács.

**Visualization:** Lilla Magyari.

**Writing – original draft:** Lilla Magyari, Csaba Pléh, Bálint Forgács.

**Writing – review & editing:** Lilla Magyari, Csaba Pléh, Bálint Forgács.

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
