## [Decision Letter · Decision Letter 0]

23 May 2022

PONE-D-22-05588The Hungarian Hubris SyndromePLOS ONE

Dear Dr. Magyari,

Thank you for submitting your manuscript to PLOS ONE. After careful consideration, we feel that it has merit but does not fully meet PLOS ONE’s publication criteria as it currently stands. Therefore, we invite you to submit a revised version of the manuscript that addresses the points raised during the review process.

We look forward to receiving your revised manuscript.

Kind regards,

Andrea Fronzetti Colladon, Ph.D.

Academic Editor

PLOS ONE

Journal Requirements:

Additional Editor Comments (if provided):

We have received the reports from our advisors on your manuscript and read them carefully. We think the reviewers provided very good assessments and recommendations. They have also raised some important concerns in regard to presentation of the paper contribution. Based on the advice received, we feel that your manuscript could be reconsidered for publication should you be prepared to incorporate revisions.

You should appreciate that this is a *major* revision. Minor changes to your manuscript will not be acceptable. Furthermore, the current decision does not guarantee an eventual acceptance of your paper - thus, the importance of your revisions.

Reviewers' comments:

Reviewer's Responses to Questions

**Comments to the Author**

1. Is the manuscript technically sound, and do the data support the conclusions?

Reviewer #1: Yes

Reviewer #2: Partly

Reviewer #3: Yes

2. Has the statistical analysis been performed appropriately and rigorously? 

Reviewer #1: Yes

Reviewer #2: Yes

Reviewer #3: Yes

3. Have the authors made all data underlying the findings in their manuscript fully available?

Reviewer #1: Yes

Reviewer #2: Yes

Reviewer #3: Yes

4. Is the manuscript presented in an intelligible fashion and written in standard English?

Reviewer #1: Yes

Reviewer #2: Yes

Reviewer #3: Yes

5. Review Comments to the Author

Reviewer #1: This study is clearly reported and warrants publication. It makes the case of proving the hypothesis of Hubris syndrome among the re-elected Hungarian prime ministers, especially in the case of Orbán. Some of the remarks on the spontaneous speech and discussions on the political meanings of the syndrome appear superficial for an interactional and discourse analysis point of view, thus the authors could make the paper more robust by remaining in the psycholinguistic paradigm. Among others, parliamentary replies are a special genre with a long history (studied by e.g. rhetorics), and the PM's usually have received training for it, too, since they are often mediatized as well. Conclusions on the political dimensions would require a more critical discourse analysis of the particular speeches and their contextualized interpretations and the role of the first plural in them.

Reviewer #2: This paper reports the results of a text analysis of Hungarian PMs over the period 1998-2018, which focuses on the usage of pronoun and verb conjugation usage by these PMs in spontaneous parliamentary speeches. The authors posit that increased usage of these pronouns and conjugations corresponds to increased levels of “hubris”. The results suggest that two PMs stand out: Gyurcsany and Orban and that, generally speaking, both these PMs have higher WE:I ratios after re-election.

There is a lot to like about this paper. The text analysis is straightforward and the statistical analysis is coherent and appropriate. However, I have reservations about the inherent value of the study as it currently stands as well as the theoretical framework.

Starting first from the theory, it is not clear from the text why we should believe that an increased WE:I ratio in PM utterances in parliament should necessarily correspond to increased levels of “hubris”. The paper fails to offer alternative explanations for such usage, and it is reasonable to expect that such alternatives exist. Both Gyurcsany and Orban are controversial political figures in their own respects, with the former resigning from his position in 2009 and the latter espousing anti-LGTB, nationalistic, and Euroskeptic positions in his second and third terms. The current paper does not consider the possibility, for example, that the usage of plural pronouns and verb conjugations could be the result of rhetorical strategies to share responsibility (see, e.g., Håkansson, 2012). Further, there could be multiple factors at play, with rhetorical strategies used as a form of communicating the actors’ ideology as in the case of Orban. It is well known that post-2010 Orban has a radically different ideological disposition with that of first term Orban. The usage of singular versus plural pronouns and verb conujugations clearly maps onto the “us-versus-them” rhetoric employed by political populist leaders, such as Victor Orban. By referring to “we”, the populist leader is illustrating the in-group (i.e. the Hungarian “people”) and by “them/they” he would be referencing the out-group (e.g. elites). The current paper does not consider this alternative explanation for the increased usage of “we” by now-populist Victor Orban, even though this is a well-studied issue among scholars of populist rhetoric (see, e.g. Tyrkkö, 2016). The study must be able to take into account these alternative explanations in the analysis, if we are to believe that the observed increase in the WE:I ratio is directly attributable to a “hubris syndrome”

Furthermore, even if we are to believe that Orban, for instance, has indeed succumbed to an increase in hubris and that his rhetoric is an indicator of this, the study does not provide a solid discussion of the implications of these findings. Why should we care if Orban is now exhibiting features of increased hubris? Does this offer a new insight into his decisionmaking processes? Would other EU leaders learn anything new from this analysis which might assist them when dealing with him? Are there implications for voters in Hungary? More work needs to go into the policy implications of these findings.

References:

Håkansson, J. (2012). The Use of Personal Pronouns in Political Speeches: A comparative study of the pronominal choices of two American presidents.

Tyrkkö, J. (2016). Looking for rhetorical thresholds: Pronoun frequencies in political speeches. Studies in Variation, Contacts and Change in English, 17.

Reviewer #3: Recommendation: Minor Revision

1. Comments to Authors

1.1. Overview and general recommendation

1.1.1. The authors analysed and discussed the occurrence of symptoms of the Hubris Syndrome (HS) for Hungarian Prime Ministers (PMs), through the analysis of spontaneous speeches (e.g., shift from first singular person “I” to first plural person “we”).

1.1.2. The paper is fluent, technically sound and interesting to be read. However, few improvements can be still made before publication in PLOS ONE journal, which I highlighted as follows. Hence, I suggest a minor revision of the paper.

1.2. Major comments

1.2.1. I believe the authors should better explain the added value of their contribution with respect to the current scientific literature. Indeed, though the contribution of the research work clearly appears, it is not appropriately discussed in the introduction section.

1.2.2. Moreover, has any other research work investigated the Hubris Syndrome for Hungarian PMs or for other countries’ PMs? If not, please explicitly highlight this absence in the section.

1.2.3. In the Introduction section you mentioned: “Our hypotheses are the following: Orbán, due to hubristic personality trait uses the personal pronoun “we” in a higher proportion relative to “I” (WE:I ratio) in his speech than other former Hungarian PMs from his second premiership onwards.”: could you please elaborate and discuss more in depth your hypotheses related to Orbán?

1.2.4. In the Conclusion section, you mentioned that “Two of the symptoms of hubris refer to such language use: 1) when one uses the “royal we” and 2) when own interests or desires are identified with a nation’s (or organization’s) needs (Owen, 2008; Owen & Davidson, 2009). There are such typical examples of the usage of “we” in Orbán’s speeches when he speaks not only in the name of his party or the government but in the name of his voters.”: it may be interesting to evaluate the occurrence of both symptoms and perform an additional analysis concerning the differences between those two categories of symptoms. Indeed, is there any specific pattern observed for the different PMs considering such language usage?

1.3. Minor comments

1.3.1. At lines 63-64 you mentioned that “[..] (even though some positive consequences have been pointed out recently; Zeitoun et al., 2019).”: please modify it as “[..], even though some positive consequences have been pointed out recently (Zeitoun et al., 2019).”

6. PLOS authors have the option to publish the peer review history of their article (what does this mean?). If published, this will include your full peer review and any attached files.

Reviewer #1: No

Reviewer #2: No

Reviewer #3: **Yes: **Sebastiano Di Luozzo

---

## [Author Response · Author response to Decision Letter 0]

14 Jul 2022

Dear Dr. Andrea Fronzetti Colladon,

We thank the reviewers for providing us with highly valuable feedback and suggestions to improve our manuscript. We implemented major changes in our manuscript by revising our Abstract, Introduction and Discussion. In the revised manuscript, we also moved Table 1 from the Materials and Method section to the Results as it contains descriptive statistics. We also included a new analysis with a new figure (Fig 2) in our Results section. We also updated the manuscript according to PLOS ONE’s style requirements. In the marked-up copy of our manuscript we did not mark changes in style if those did not affect the text or content of the manuscript. Below we reply in each specific comment in turn.

Response to Reviewer 1

Reviewer #1: This study is clearly reported and warrants publication. It makes the case of proving the hypothesis of Hubris syndrome among the re-elected Hungarian prime ministers, especially in the case of Orbán. Some of the remarks on the spontaneous speech and discussions on the political meanings of the syndrome appear superficial for an interactional and discourse analysis point of view, thus the authors could make the paper more robust by remaining in the psycholinguistic paradigm. 

Authors: We would like to thank the Reviewer for the careful evaluation of our paper. We are grateful for pointing out that we did not discuss interactional and discourse analysis studies of political speeches in the earlier version of our manuscript. In our study, we focused on presumably spontaneous speeches of politicians (replies, re-replies, remarks during debates) which are very different from pre-written, classical political speeches typically analyzed in other studies. Our research methods and questions are also rather far from classical discourse analysis. The main focus of our paper is a quantitative analysis of specific, focused linguistic markers, instead a content driven and/or comprehensive descriptive analysis of political discourse. We analyzed a particular linguistic marker, ratio of WE:I in our study which has been related exclusively to hubristic behaviour in the literature so far. However, we agree that studies of discourse analysis provide very important insight on the possible function of “we” in political speeches. Therefore, we included a discussion of such studies in the revised Discussion (5-6 paragraphs, p 15.).

Reviewer #1: Among others, parliamentary replies are a special genre with a long history (studied by e.g. rhetorics), and the PM's usually have received training for it, too, since they are often mediatized as well. 

Authors: We also agree with the reviewer that parliamentary responses may reflect training in rhetoric. However, we would like to point out that rhetoric training may focus on a number of factors, but we are not aware of any rhetorical school or approach that would emphasize (and successfully implement) the enhanced use of WE over I in relatively spontaneous responses. Should the Reviewer have any particular author in mind, we are ready to consider the idea in further depth. One may of course still assume that the WE:I ratio changed due to rhetoric training, but then such training should have happened systematically prior the reelection of each PM. We believe that such an explanation of our results is much less likely than being under the influence of Hubris Syndrome (HS), as changes in WE:I ratio has been associated with this syndrome.

Reviewer #1: Conclusions on the political dimensions would require a more critical discourse analysis of the particular speeches and their contextualized interpretations and the role of the first plural in them.

Authors: We completely agree that the role of “we” in the context of the speeches could be better assessed by a qualitative analysis which is out of the scope of our study. At the same time, our aim in the current paper was a rather different one. Instead of providing an in-depth analysis of content and context of political speeches and discourse in the Hungarian Parliament, our focus has been on a particular quantitative linguistic marker, associated with hubristic behavior in prior research. In other words, while the role of the first plural may be of particular interest of additional analyses indeed, it would go in a quite different direction we set out to investigate. In the updated version of our manuscript, we nevertheless added in the Discussion that “Future qualitative studies would be necessary to reveal the pragmatic function and referent of “we” in political speeches and whether there is a change also in their function accompanying the change in the WE:I ratio” (p 16, lines 346-348). 

We also slightly modified our Abstract and the Discussion to moderate our conclusion (e.g., we changed “The results (…) confirm the hypothesis” to “This finding suggests” in the Abstract (line 31) and we changed “Our results demonstrate” to “Our results suggest” in the last paragraph of the Discussion (line 361)). 

Response to Reviewer 2

Reviewer #2: This paper reports the results of a text analysis of Hungarian PMs over the period 1998-2018, which focuses on the usage of pronoun and verb conjugation usage by these PMs in spontaneous parliamentary speeches. The authors posit that increased usage of these pronouns and conjugations corresponds to increased levels of “hubris”. The results suggest that two PMs stand out: Gyurcsany and Orban and that, generally speaking, both these PMs have higher WE:I ratios after re-election.

There is a lot to like about this paper. The text analysis is straightforward and the statistical analysis is coherent and appropriate. However, I have reservations about the inherent value of the study as it currently stands as well as the theoretical framework.

Starting first from the theory, it is not clear from the text why we should believe that an increased WE:I ratio in PM utterances in parliament should necessarily correspond to increased levels of “hubris”. The paper fails to offer alternative explanations for such usage, and it is reasonable to expect that such alternatives exist. 

Authors: We thank the Reviewer for the valuable suggestions to improve our manuscript. Following the Reviewer’s suggestion, we included additional alternative explanations in our text (5-6 paragraphs of the revised Discussion, p 15.). More specifically, we discuss studies that relate the frequent usage of “we” to rhetorical strategies (even though they do not address the question of spontaneous and pre-prepared political speeches). Importantly, we would like to point out that by comparing Hungarian PMs to one another is expected to mitigate such differences: opportunities and/or requirements for rhetorical training were likely to be comparable among them (even though there is no information in the public record in this matter in Hungary). We are of course very much open to any additional alternative explanations the Reviewer may have in mind, and we are ready to consider and potentially address them in our manuscript. Further on, we still believe that the explanation included in the first version of our paper, that high-status individual might use more “we” because of their social skills and leadership position, is one of the most powerful alternative accounts (p 14, lines 311-317). The latter explanation, nevertheless has limited explanatory power regarding our data, because all analyzed individuals were equally top ranking MPs, therefore, status is insufficient to account for the relative difference in the WE:I ratios among them. 

It is true that an increased WE:I ratio does not necessarily and exclusively correspond to HS. However, we also would like to point out that a change in the WE:I ratio has been linked to hubristic behaviour in the literature, via both quantitative and qualitative analyses of language use, behavior and political and business decision making. Importantly such a pattern has been observed in spontaneous (as opposed to pre-prepared) speech, which is a crucial detail: such a change of speech pattern seems to be very likely not under the deliberate control of the speaker. 

However, we indeed measured only WE:I ratio and did not analyze specific behavioral patterns, therefore, we are not in the position to establish a causal link. Our prediction, nevertheless, and a hypothesis driven research question was whether the WE:I ratio change in accordance with obtaining and experiencing high political power. To reflect on this correlation, we slightly modified our Abstract and the Discussion to moderate our conclusion (e.g., we changed “The results (…) confirm the hypothesis” to “This finding suggests” in the Abstract (line 31) and we changed “Our results demonstrate” to “Our results suggest” in the last paragraph of the Discussion (line 361)). 

Reviewer #2: Both Gyurcsany and Orban are controversial political figures in their own respects, with the former resigning from his position in 2009 and the latter espousing anti-LGTB, nationalistic, and Euroskeptic positions in his second and third terms. The current paper does not consider the possibility, for example, that the usage of plural pronouns and verb conjugations could be the result of rhetorical strategies to share responsibility (see, e.g., Håkansson, 2012).

Authors: We are grateful for the useful reference the Reviewer recommended. We also acknowledge in the revised Discussion that qualitative analysis could give insight about the exact role and function of “we” in our speech samples (lines 346-348). Along with Håkansson’s, we also refer to a study by Karapetjana (2011) which describes the function of exclusive “we” as sharing responsibility in political interviews (line 320). 

At the same time, aim of the current paper is not a qualitative analysis of the content of political debates and messaging, but a quantitative study of a linguistic marker that may be telling of a change in the overall psychological functioning of political leaders in general.

Reviewer #2: Further, there could be multiple factors at play, with rhetorical strategies used as a form of communicating the actors’ ideology as in the case of Orban. It is well known that post-2010 Orban has a radically different ideological disposition with that of first term Orban. The usage of singular versus plural pronouns and verb conujugations clearly maps onto the “us-versus-them” rhetoric employed by political populist leaders, such as Victor Orban. By referring to “we”, the populist leader is illustrating the in-group (i.e. the Hungarian “people”) and by “them/they” he would be referencing the out-group (e.g. elites). The current paper does not consider this alternative explanation for the increased usage of “we” by now-populist Victor Orban, even though this is a well-studied issue among scholars of populist rhetoric (see, e.g. Tyrkkö, 2016). The study must be able to take into account these alternative explanations in the analysis, if we are to believe that the observed increase in the WE:I ratio is directly attributable to a “hubris syndrome”.

Authors: Thank you for the suggestions and we are grateful for the suggested literature on this topic. In the revised Discussion, we refer to Trykkö (2016) and van Dijk (2006) (p 15). We, however, also note that Trykkö (2016) and Håkansson (2012) focused on pre-written, not spontaneous speeches. Both found that the first-person plural (“we”) was the most frequent among other types of personal pronouns, and Trykkö also showed that there is a tendency for using increasingly more “we” in political speeches from 1920. However, rhetorical strategies may play a stronger role in such speeches, while our study aimed at spontaneous speeches, exactly because of the prediction in the literature that uncontrollable alterations of speech patterns may emerge due to HS.

Regarding the expression of populist ideological polarization by ingroups and outgroups, the above works imply that both the first-person plurals and also the third-person plural would increase. Following up on this idea, we have carried an analysis of the occurrence frequencies of all pronouns in the speech samples we analyzed and found that for all politicians, in each parliamentary cycles, the second-person pronouns (“you”) were the most frequent ones in the selected spontaneous speech samples. This finding further corroborates that our speech samples might be different from classical political speeches in that they are born out of heated parliamentary debates, spontaneously (please, see the first paragraph if the revised Results section on p 8-9, Fig 2 on p 9, and the sixth paragraph of the revised Discussion on p 15). Moreover, we also found that while the frequency of “we” increased for both Orbán and Gyurcsány from their first premiership to the second, the frequency of „they” did not change even in ther case (please see Supporting information, Supplementary Results). We included and discussed these results in the revised manuscript and Supporting information. 

In the context of populist rhetoric, it should be noted, however, that even the pronoun “we” could be used either in an exclusive (“we” the Fidesz party and its followers) or inclusive manner (“we”, all the Hungarians), and Orbán strategically uses the inclusive “we” in an exclusive manner (“we”, the Fidesz are the real and only Hungarians, the rest are traitors). In other words, identifying exclusive vs. inclusive “we” (and the referent of “we” in general) is in and by an interpretive and highly challenging question, where it is often not clear what matters more, the intention of the speaker or the interpretation by the audience. Our analyses were not aimed to capture such fine-grained nuances, as they require qualitative speech analysis of each “we” uttered. 

Finally, we would also like to point out that, in our view, Orbán was a nationalist populist prior to 2010 as well, already during his first term, from 1998 to 2002. Our hypotheses were driven by the fact that he obtained excessive, complete power in Hungary in 2010 (a supermajority with the power to change the constitution, which he did), which, in turn, may have made him more prone to express hubristic traits, including a change in his spontaneous speech pattern. It is true that Orbán was considered a moderate conservative in the West prior to 2010, but in fact he has borrowed skillfully and unabashedly from the far right even back then, such as utilizing national symbols (“kokárda” or the national flag) for political purposes during elections, and flirted openly with the legacy of Miklós Horthy, the autocratic leader of Hungary in the interwar period (which remained largely underreported and labelled mostly as “leftist” concerns). It is also true that in the late 1980s, Orbán’s political party, Fidesz was indeed affiliated with liberal forces, but this episode ended after the 1994 elections, when Orbán decided to fill the power vacuum appearing on the political right. Although in our original manuscript we have already mentioned supermajority in the Discussion (lines 351-353), we also refer to the supermajority now in the revised Introduction (lines 78, 96, 102).

Reviewer #2: Furthermore, even if we are to believe that Orban, for instance, has indeed succumbed to an increase in hubris and that his rhetoric is an indicator of this, the study does not provide a solid discussion of the implications of these findings. Why should we care if Orban is now exhibiting features of increased hubris? Does this offer a new insight into his decision making processes? Would other EU leaders learn anything new from this analysis which might assist them when dealing with him? Are there implications for voters in Hungary? More work needs to go into the policy implications of these findings.

Authors: Thank you for raising this crucial issue. We agree with the Reviewer that political implications could be spelled out in greater detail. However, our main goal is to analyze speech patterns that may be telling of a change of personality that my lead to bad decision making in general. Therefore, we would like to draw more general conclusions instead of a warning about Orbán for the EU. Populist or not, hubristic behavior leads to bad governance and ill-suited political and economic decisions, which Orbán may exemplify well, but our main point is that the deterioration of leadership abilities may be detected in linguistic markers of spontaneous speech. Along these lines, we extended now the revised Discussion by emphasizing that it is important to study hubris in the case of powerful politicians, because it might also lead to political misjudgments and even to unethical behaviour (p 16, lines 358-360). HS theory does not provide insight into the mechanisms of the deterioration of decision making, but the matter is certainly worth future exploration. 

Reviewer #2:

References:

Håkansson, J. (2012). The Use of Personal Pronouns in Political Speeches: A comparative study of the pronominal

choices of two American presidents.

Tyrkkö, J. (2016). Looking for rhetorical thresholds: Pronoun frequencies in political speeches. Studies in Variation,

Contacts and Change in English, 17.

Authors: We thank you for the references.

Response to Reviewer 3

Reviewer #3: Recommendation: Minor Revision

1. Comments to Authors

1.1. Overview and general recommendation

1.1.1. The authors analysed and discussed the occurrence of symptoms of the Hubris Syndrome (HS) for Hungarian Prime Ministers (PMs), through the analysis of spontaneous speeches (e.g., shift from first singular person “I” to first plural person “we”).

1.1.2. The paper is fluent, technically sound and interesting to be read. However, few improvements can be still made before publication in PLOS ONE journal, which I highlighted as follows. Hence, I suggest a minor revision of the paper.

1.2. Major comments

1.2.1. I believe the authors should better explain the added value of their contribution with respect to the current scientific literature. Indeed, though the contribution of the research work clearly appears, it is not appropriately discussed in the introduction section.

Authors: We are grateful for the suggestions. We extended the Introduction of the revised manuscript with emphasizing the gap which is filled by our study. We explain that there are only a few studies dealing hubristic behaviour in politicians, although hubristic behaviour can lead to bad decisions and has a risk for failure. Moreover, linguistic markers of hubris have been described in English, hence, this study also gives insight about linguistic markers in a very different (non-Indo-European) language, Hungarian (p 4, lines 68-72). 

 In the revised Discussion, we emphasize again that hubristic traits in leaders might lead to bad decisions and unethical behaviour, hence, it is extremely important to study hubris in powerful politicians’ behaviours (p 16, lines 358-360).

Reviewer #3: 1.2.2. Moreover, has any other research work investigated the Hubris Syndrome for Hungarian PMs or for other countries’ PMs? If not, please explicitly highlight this absence in the section.

Authors: To the best of our knowledge, no other study investigated hubris in Hungarian PMs. We also did not find studies investigating PMs in other countries in the English language international journals, except for those two that inspired our own research, studying hubris in British PMs (Garrard et al., 2014), and comparing British PMs and US Presidents (Owen, Davidson, 2009). In the revised Introduction, we now make a stronger emphasis of the lack of deeper research into hubristic behaviour of politicians (p 4, lines 68-70). 

Reviewer #3: 1.2.3. In the Introduction section you mentioned: “Our hypotheses are the following: Orbán, due to hubristic personality trait uses the personal pronoun “we” in a higher proportion relative to “I” (WE:I ratio) in his speech than other former Hungarian PMs from his second premiership onwards.”: could you please elaborate and discuss more in depth your hypotheses related to Orbán?

Authors: We added more explanation to the hypothesis in the revised Introduction, thank you for pointing out this matter (p 5, lines 95-102).

Reviewer #3: 1.2.4. In the Conclusion section, you mentioned that “Two of the symptoms of hubris refer to such language use: 1) when one uses the “royal we” and 2) when own interests or desires are identified with a nation’s (or organization’s) needs (Owen, 2008; Owen & Davidson, 2009). There are such typical examples of the usage of “we” in Orbán’s speeches when he speaks not only in the name of his party or the government but in the name of his voters.”: it may be interesting to evaluate the occurrence of both symptoms and perform an additional analysis concerning the differences between those two categories of symptoms. Indeed, is there any specific pattern observed for the different PMs considering such language usage?

Authors: We agree that it would be very interesting to evaluate the role of “we” in our samples. However, answering such a question requires quantitative content-analysis, which lays outside of the scope of our current question and quantitative approach. We agree with the Reviewer that this is an important question and it would be important to further specify the role of “we” once it becomes proportionately more frequent in spontaneous speech of high-power politicians. Therefore, we included in the revised Discussion that future qualitative studies reveal the role of “we” in PMs spontaneous political speeches (lines 346-348). Please also see our response above to Reviewer 2, who raised similar points.

Reviewer #3: 1.3. Minor comments

1.3.1. At lines 63-64 you mentioned that “[..] (even though some positive consequences have been pointed out recently; Zeitoun et al., 2019).”: please modify it as “[..], even though some positive consequences have beenpointed out recently (Zeitoun et al., 2019).”

Authors: We corrected this sentence, thank you.

---

## [Decision Letter · Decision Letter 1]

5 Aug 2022

The Hungarian hubris syndrome

PONE-D-22-05588R1

Dear Dr. Magyari,

We’re pleased to inform you that your manuscript has been judged scientifically suitable for publication and will be formally accepted for publication once it meets all outstanding technical requirements.

We ask that you copyedit the manuscript thoroughly before submitting your final version. In particular, we ask that you amend some more colloquial phrases in the abstract and introduction, such as "getting drunk with power".

Kind regards,

Hanna Landenmark

Staff Editor

PLOS ONE

Additional Editor Comments (optional):

Reviewers' comments:

Reviewer's Responses to Questions

**Comments to the Author**

1. If the authors have adequately addressed your comments raised in a previous round of review and you feel that this manuscript is now acceptable for publication, you may indicate that here to bypass the “Comments to the Author” section, enter your conflict of interest statement in the “Confidential to Editor” section, and submit your "Accept" recommendation.

Reviewer #1: All comments have been addressed

Reviewer #2: All comments have been addressed

Reviewer #3: All comments have been addressed

2. Is the manuscript technically sound, and do the data support the conclusions?

Reviewer #1: Yes

Reviewer #2: Yes

Reviewer #3: Yes

3. Has the statistical analysis been performed appropriately and rigorously? 

Reviewer #1: Yes

Reviewer #2: Yes

Reviewer #3: Yes

4. Have the authors made all data underlying the findings in their manuscript fully available?

Reviewer #1: Yes

Reviewer #2: Yes

Reviewer #3: Yes

5. Is the manuscript presented in an intelligible fashion and written in standard English?

Reviewer #1: Yes

Reviewer #2: Yes

Reviewer #3: Yes

6. Review Comments to the Author

Reviewer #1: my concerns have now been addressed and the revised draft is publishable now. I congratulate the authors for the rigorous analysis of data.

Reviewer #2: I wish to thank the authors for addressing my concerns and suggestions. I am satisfied with the revisions.

Reviewer #3: The authors addressed all the reviewers' comments, which improved the research manuscript. For this reason, I feel that the manuscript is now suitable for publication in PLOS One.

7. PLOS authors have the option to publish the peer review history of their article (what does this mean?). If published, this will include your full peer review and any attached files.

Reviewer #1: **Yes: **Petteri Laihonen

Reviewer #2: No

Reviewer #3: **Yes: **Sebastiano Di Luozzo

---

## [Editor Report · Acceptance letter]

15 Aug 2022

PONE-D-22-05588R1 

The Hungarian hubris syndrome 

Dear Dr. Magyari:

I'm pleased to inform you that your manuscript has been deemed suitable for publication in PLOS ONE. Congratulations! Your manuscript is now with our production department. 

Kind regards, 

on behalf of

Dr. Hanna Landenmark 

Staff Editor

PLOS ONE